# Complete morphologies of basal forebrain cholinergic neurons in the mouse

**Hao Wu[1], John Williams[1], Jeremy Nathans[1,2,3]\***

[1]Department of Molecular Biology and Genetics, Howard Hughes Medical Institute, Johns Hopkins University School of Medicine, Baltimore, United States; [2]Department of Neuroscience, Johns Hopkins University School of Medicine, Baltimore, United States; [3]Department of Opthalmology, Johns Hopkins University School of Medicine, Baltimore, United States

**Abstract** The basal forebrain cholinergic system modulates neuronal excitability and vascular tone throughout the cerebral cortex and hippocampus. This system is severely affected in Alzheimer's disease (AD), and drug treatment to enhance cholinergic signaling is widely used as symptomatic therapy in AD. Defining the full morphologies of individual basal forebrain cholinergic neurons has, until now, been technically beyond reach due to their large axon arbor sizes. Using genetically-directed sparse labeling, we have characterized the complete morphologies of basal forebrain cholinergic neurons in the mouse. Individual arbors were observed to span multiple cortical columns, and to have >1000 branch points and total axon lengths up to 50 cm. In an AD model, cholinergic axons were slowly lost and there was an accumulation of axon-derived material in discrete puncta. Calculations based on published morphometric data indicate that basal forebrain cholinergic neurons in humans have a mean axon length of ~100 meters.

**\*For correspondence:** jnathans@jhmi.edu

**Reviewing editor**: Franck Polleux, Columbia University, United States

## Introduction

The mammalian cerebral cortex and hippocampus are densely innervated by cholinergic fibers that originate in the basal forebrain (*Mesulam, 2004*). Most, if not all, cortical and hippocampal neurons respond to cholinergic signals using muscarinic and/or nicotinic acetylcholine receptors localized to pre- and/or postsynaptic sites. Activation of postsynaptic nicotinic receptors leads to transient depolarizing currents with a high calcium:sodium ratio and activation of postsynaptic muscarinic receptors leads to a sustained reduction in potassium currents, both effects producing a general elevation in excitability (*Lucas-Meunier et al., 2003*). Additionally, cholinergic activation of muscarinic receptors in the microvasculature leads to the production of nitric oxide, producing vasodilation (*Hamel, 2004*).

A wide variety of experiments in primates, cats, and rodents have implicated cholinergic neurotransmission from the basal forebrain to the cortex and hippocampus in attention, memory, and plasticity. In cat primary visual cortex, cholinergic input enhances neuronal responses to a preferred visual stimulus (*Sato et al., 1987*), and in rat motor cortex, cholinergic input promotes more complex motor sequences in response to electrical stimulation (*Berg et al., 2005*). In rat barrel cortex and auditory cortex, cholinergic input is required for experience-dependent synaptic plasticity and circuit reorganization (*Baskerville et al., 1997*; *Kilgard and Merzenich, 1998*; *Zhu and Waite, 1998*). In contrast to the high spatial precision of glutamatergic and GABAergic neurotransmission, current evidence indicates that the basal forebrain cholinergic system modulates neuronal excitability and vascular tone over large target areas.

The basal forebrain cholinergic system is of special interest because it degenerates in a variety of common neurologic diseases, including Alzheimer's disease (AD) and Parkinson's disease, to an extent

**eLife digest** The human brain is made up of roughly 80 to 100 billion neurons, organized into extensive networks. Each neuron consists of a number of components: a cell body, which contains the nucleus; numerous short protrusions from the cell body called dendrites; and a long thin structure called an axon that carries the electrical signals generated in the cell body and the dendrites to the next neuron in the network.

One of the most studied networks in the human brain is the basal forebrain network, which is made up of large neurons that communicate with one another using a chemical transmitter called acetylcholine. This network has a key role in cognition, and its neurons are among the first to degenerate in Alzheimer's disease. However, relatively little is known about the structure of these 'cholinergic' neurons because their large size makes them difficult to study using standard techniques.

Now, Wu et al. have visualized, for the first time, the complete 3D structure of cholinergic neurons in the mouse forebrain. The mice in question had been genetically modified so that only ten or so of their many thousands of cholinergic neurons expressed a distinctive 'marker' protein. This made it possible to distinguish these neurons from surrounding brain tissue in order to visualize their structures. The resulting pictures clearly illustrate the neurons' complexity, with individual axons in adult mice displaying up to 1000 branches.

Measurements showed that each cholinergic axon in the mouse brain is roughly 30 centimeters long, even though the brain itself is less than 2 centimeters from front to back. Based on measurements by other researchers, Wu et al. calculated that the axons of single cholinergic neurons in the human brain are about 100 meters long on average.

The extreme length and complex branching structure of cholinergic forebrain neurons helps to explain why each neuron is able to modulate the activity of many others in the network. It could also explain their vulnerability to degeneration, as the need to transport materials over such long distances may limit the ability of these neurons to respond to damage.

that correlates with the severity of dementia (*Schliebs and Arendt, 2011*). In advanced AD, the relative loss of cholinergic innervation varies by region, with the temporal lobe showing the greatest loss of fibers and the primary sensory, motor, and anterior cingulate cortices showing the least loss (*Geula and Mesulam, 1989*). The relative extent of cholinergic fiber loss in different cortical areas appears to correlate inversely with fiber density in the normal brain, suggesting that disease-associated fiber loss progresses at roughly equal rates throughout the cortex and that those regions that began with the fewest fibers are the first to become denuded of cholinergic input. The loss of forebrain cholinergic innervation in AD has stimulated the development of pharmacotherapy to enhance cholinergic signaling as an approach to partially ameliorate cognitive symptoms (*Burns et al., 2006*).

From the preceding paragraphs it is clear that an accurate anatomic description of the basal forebrain cholinergic system is important for understanding its function and its susceptibility to degeneration. At present, this description, which derives from retrograde and anterograde filling and from histochemical and immunohistochemical staining, provides a view that is accurate as a statistical picture but is incomplete in one critical respect: the morphologies of individual cholinergic axon arbors are unknown because their extraordinarily large size has, thus far, precluded classical tracer filling and reconstruction. In earlier work, we demonstrated the utility of extremely sparse *CreER/loxP* labeling methods for visualizing axonal and dendritic morphologies of large neurons, including forebrain cholinergic neurons (*Rotolo et al., 2008*; *Badea et al., 2009*). In the present work, we use this approach to visualize and quantify the full 3-dimensional axonal morphologies of individual forebrain cholinergic neurons and to define changes in these arbors in response to disease progression in a mouse model of AD.

## Results

### A genetic system for extremely sparse labeling of cholinergic neurons

In earlier work, we generated an *IRES-CreER* knock-in in the 3′ untranslated region of the gene coding for choline acetyl transferase (ChAT; *Rotolo et al., 2008*). This allele expresses relatively low levels of

CreER and, consequently, shows no recombination of Cre-activated reporters in the absence of tamoxifen or 4-hydroxytamoxifen (4HT), a prerequisite for visualizing genetically marked neurons at densities of <10 labeled neurons per brain. To visualize large axon arbors in their entirety and to efficiently survey dozens of brains, we chose the highly sensitive histochemical reporter human placental alkaline phosphatase (AP), a GPI-anchored protein that distributes relatively uniformly along dendrites and axons (*Rotolo et al., 2008*). AP histochemistry works efficiently with relatively thick (300 μm) vibratome sections, which minimizes the number of sections required per brain and thereby simplifies the logistics of staining, imaging, and tracing. To minimize background reporter activity, we used an AP reporter knock-in at the *Gt(ROSA)26Sor* locus (referred to as *R26*) in which the 3′ half of the AP coding region is in reverse orientation and Cre-mediated recombination restores this segment to the correct orientation (*R26IAP*, '*I*' stands for 'inverted'; *Figure 1A*; *Badea et al., 2009*). In contrast to standard reporters that are maintained in a repressed state by a *loxP-stop-loxP* cassette, the *R26IAP* locus shows undetectable reporter activity prior to Cre-mediated recombination.

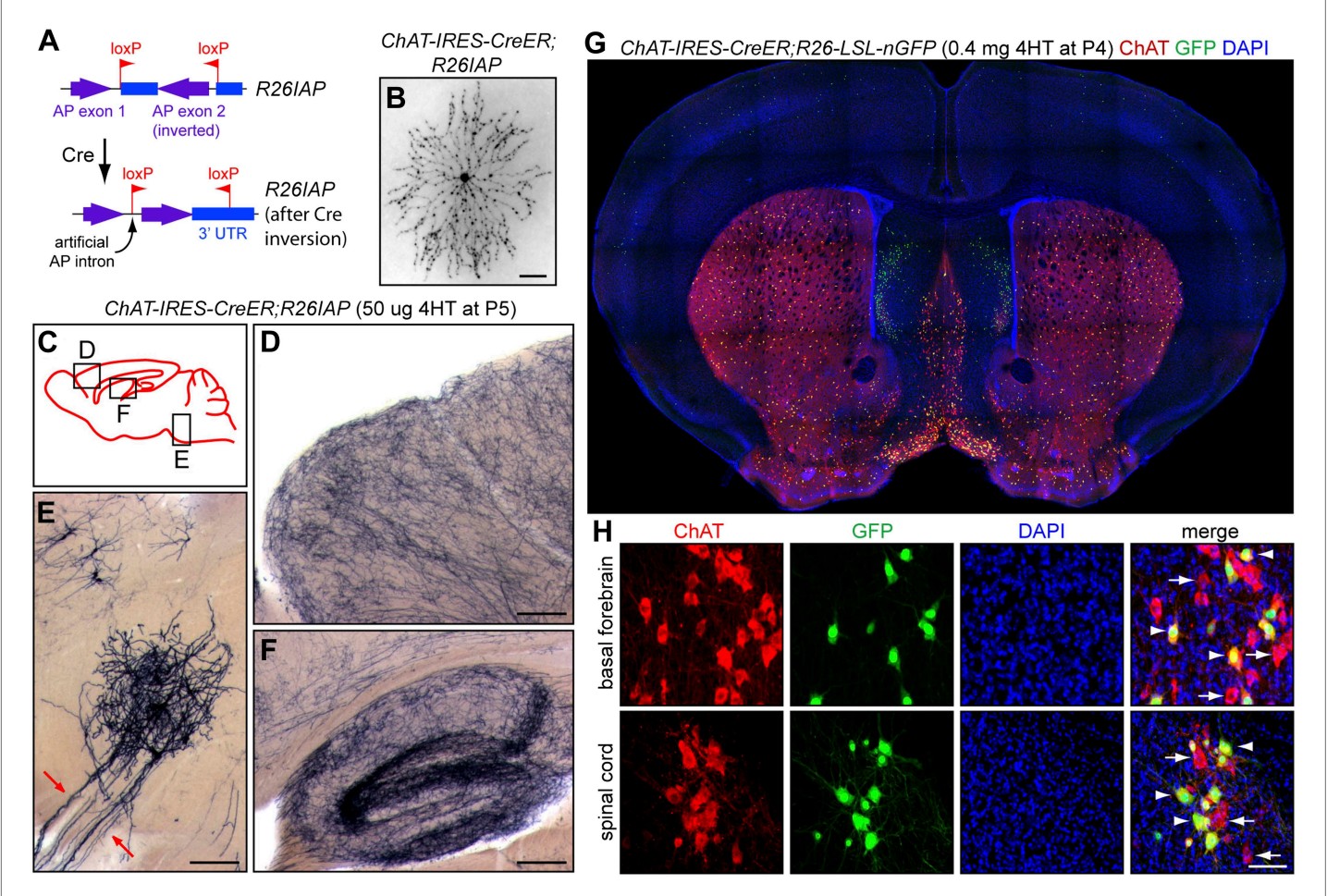

**Figure 1**. Cholinergic neuron specificity of Cre-mediated recombination. (**A**) Structure of the *R26IAP* knock-in. In the absence of Cre-mediated recombination, the 3′ half of the AP coding region is inverted in the germline configuration. It assumes the correct orientation following Cre-mediated recombination between inverted *loxP* sites. (**B**) P30 retina from *Chat-IRES-CreER;R26IAP* mice treated with 4HT. AP histochemistry labels cholinergic (starburst) amacrine cells. Scale bar, 100 μm. (**C**–**F**) P30 brain from *Chat-IRES-CreER;R26IAP* mice treated with high dose 4HT at P5. AP histochemistry labels numerous axons throughout the cortex (**D**) and hippocampus (**F**), as well as cranial motor neurons (**E**), the axons of which are seen exiting the brain stem (red arrows). Scale bars in **D**–**F**, 200 μm. (**G** and **H**) Coronal sections of P30 forebrain from *Chat-IRES-CreER;R26-LSL-nGFP* mice treated with high dose 4HT at P4. Approximately 50% of cholinergic neurons in the basal forebrain, medial septal nucleus, striatum, and spinal cord (visualized with ChAT immunohistochemistry) are GFP+. Medial to the striatum, a distinctive group of GFP+ cell is ChAT−; these cells presumably expressed *Chat* (and, therefore, *Cre*) in the early postnatal period and then repress *Chat* expression in adulthood. In (**H**), arrows point to ChAT+;GFP− neurons and arrowheads point to ChAT+; GFP+ neurons. Scale bar, 50 μm.

The specificity of the *Chat-IRES-CreER* driver has been documented by *Rotolo et al. (2008)* and *Badea et al. (2009)* and is demonstrated here with *Chat-IRES-CreER;R26IAP* mice based on reporter expression in (1) starburst amacrine cells, the only cholinergic retinal neurons (*Figure 1B*), (2) a uniform network of fibers in the cortex and hippocampus, as expected for the axon arbors of forebrain cholinergic neurons (*Figure 1C,D,F*), and (3) cranial motor neurons in the brainstem (*Figure 1E*). *Chat-IRES-CreER* activation of a nuclear localized GFP reporter (encoded by a *R26-loxP-stop-loxP-nGFP* knock-in) shows co-localization with ChAT immunoreactivity in the basal forebrain, septal nucleus, and ventral spinal cord as expected (*Figure 1G,H*). Interestingly, a small population of ChAT-negative forebrain cells, located medial to the striatum, shows GFP expression in the adult, implying that these cells transiently express the *ChAT* gene at the time of 4HT injection [postnatal day (P)4] but not at later times (*Figure 1G*).

## Morphologies of individual forebrain cholinergic neurons

A series of 4HT titration experiments with *Chat-IRES-CreER;R26IAP* mice showed that intraperitoneal (IP) injection of 1–5 µg 4HT at P4-5 resulted in ~10 forebrain cholinergic neurons labeled per brain. Using this protocol, 67 well-separated forebrain cholinergic neurons were imaged and 12 of these neurons–8 from P12 brains and 4 from P30 brains–were traced (*Figures 2, 3, 4B*, *Figure 2—figure supplement 1*). Among the traced arbors, nine were in the cortex, two were in the hippocampus, and one was in the olfactory bulb. For each of the remaining 55 neurons, collected between 1 and 12 months of age, we determined the soma location and the boundaries of the arbor territory.

At both P12 and P30, large and complex axon arbors were observed (*Figures 2 and 3*). The mean axon length per neuron was 13 cm at P12 (n = 8; range 3–42 cm) and 31 cm at P30 (n = 4; range 11–49 cm), and at both ages, the density of branches averaged 4–5 per mm of axon length (range 3–7 per mm), giving a mean of >1000 branch points per arbor at P30, including occasional branches along the axon's initial segment (*Figure 4E–G*, *Figure 2—figure supplement 1A*, *Figure 4—figure supplement 1*). In a sample of nine basal forebrain cholinergic neurons, the mean dendritic arbor lengths were 9.6 mm at P12 (n = 5 arbors) and 11.5 mm at P30 (n = 4 arbors; *Figure 2—figure supplement 1*). The territory influenced by each cholinergic arbor was estimated by enclosing the axon traces from each 300 µm tissue section with the smallest possible convex polygon, calculating the area of each polygon, and summing the resulting polygonal volumes (polygon area × 300 µm) across all of the sections populated by the arbor of interest (*Figure 4A,C*). Using this measure, individual axon density–defined as mm of axon length per $mm^3$ of polygonal volume for an individual axon—was found to vary by a factor of ~10 among traced arbors at both P12 (n = 8) and at P30 (n = 4; *Figure 4F,G*), reflecting significant variation in the size and compactness of cholinergic arbors. The comparison between P12 and P30 implies that there is substantial growth of cholinergic axons after the second week of postnatal life. Although the polygon method somewhat over-estimates the territory influenced by an arbor (as illustrated in *Figure 4A*), it provides a good measure of the linear extent of the arbor, giving a mean value of ~2 mm in the adult brain, as seen in a compilation of the mediolateral extents of forebrain cholinergic axon arbors (*Figure 4H,I*).

The coverage factor for forebrain cholinergic axon arbors—defined as the number of arbor territories that encompass any arbitrary point in the cortex and hippocampus—can be calculated based on a mean volume of 1.35 $mm^3$ for axon arbor territories at P30 (n = 4; *Figure 4F*), a total volume of 130 $mm^3$ for the adult mouse cerebral cortex (109 $mm^3$) and hippocampus (21 $mm^3$; *Kovacević et al., 2005*), and estimates of the number of forebrain cholinergic neurons of 4500 and 6632 (*Boncristiano et al., 2002*; *Perez et al., 2007*). This calculation gives a mean coverage factor of 47–69. The analogous dendrite coverage factor and the density of dendrites in the basal forebrain regions where cholinergic projection neuron cell bodies reside (the medial septal nucleus, ventral diagonal band, and horizontal limb of the diagonal band) can also be calculated based on the volume of these territories (0.667 $mm^3$; *Paxinos and Franklin, 2001*), the mean dendrite volume per cholinergic neuron at P30 (0.0694 $mm^3$; n = 4; *Figure 2—figure supplement 1*), the mean dendrite length per cholinergic neuron at P30 (11.5 mm; n = 4; *Figure 2—figure supplement 1C,D*), and the estimated number of forebrain cholinergic neurons noted above. These calculations give a dendrite coverage factor of 470–690, and a density of cholinergic dendrites in the basal forebrain of 0.078–0.114 µm/µm³.

Numerous retrograde and anterograde labeling studies have investigated the correlation between the territories targeted by forebrain cholinergic arbors and the locations of the corresponding cell bodies (e.g., *McKinney et al., 1983*; *Saper, 1984*; *Woolf et al., 1986*; *Baskerville et al., 1993*). In our

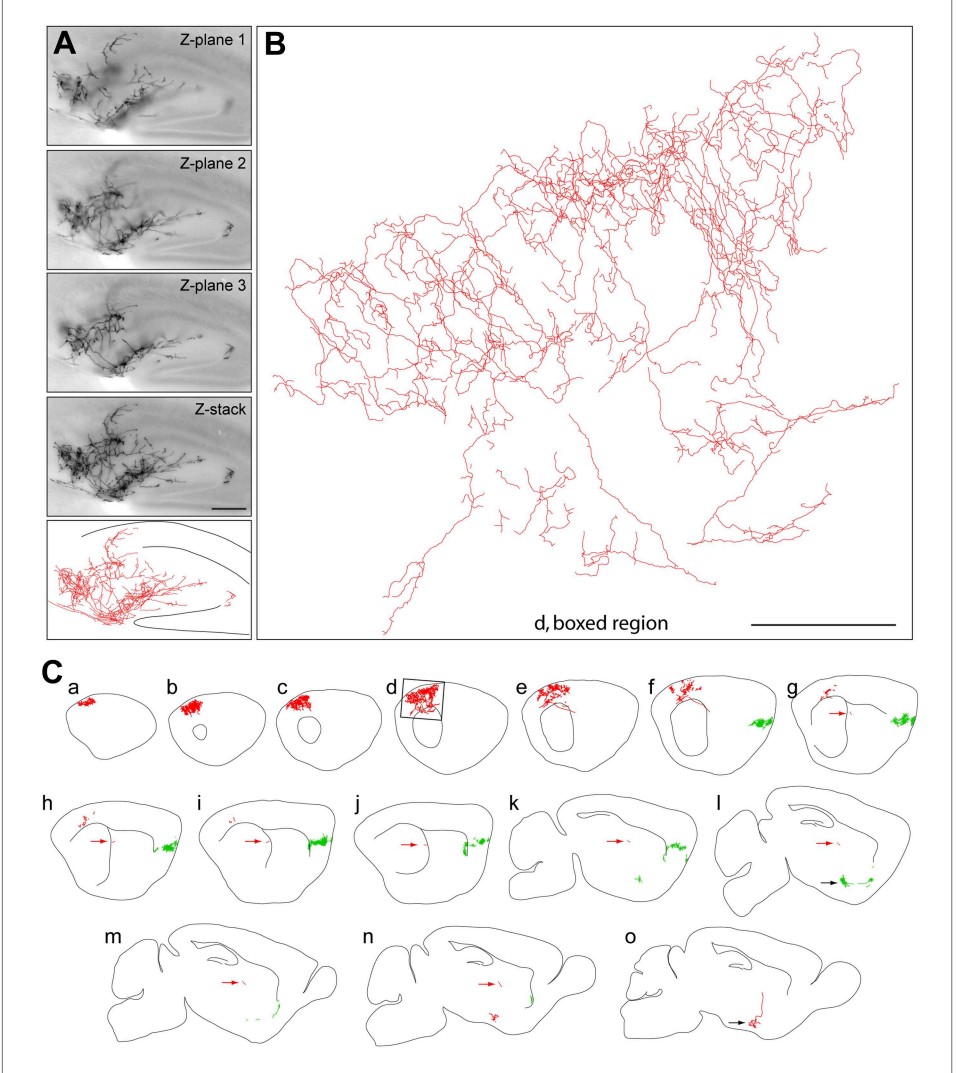

**Figure 2**. Axon arbors of forebrain cholinergic neurons from P30 *Chat-IRES-CreER;R26IAP* mice visualized with sparse Cre-mediated recombination. (**A**) Part of the arbor of a forebrain cholinergic neuron in a P30 hippocampus visualized in a single 300 μm section at three Z-planes and in a Z-stacked image. Bottom, the traced arbor. Scale bar, 200 μm. (**B** and **C**) Fifteen consecutive 300 μm sagittal sections from a single P30 hemisphere (**C**) with two fully traced AP+ forebrain cholinergic neurons, colored red and green. Black arrows in panels l and o, the two cell bodies. Red arrows in g–n, the proximal axon segment for the red neuron trace. (**B**) An enlarged view of the boxed region of section d in (**C**). Scale bar in (**B**), 500 μm (corrected for tissue shrinkage in BBBA).

The following figure supplements are available for figure 2:

**Figure supplement 1**. Dendrite structure among forebrain cholinergic neurons.

---

dataset of 67 neurons, there was a clear correlation between soma location and axon arbor position along the mediolateral axis, specifically, cell bodies located more laterally in the basal forebrain give rise to arbors that reside in more lateral cortical, hippocampal, or olfactory bulb territories, supporting the general conclusion that there is a rough topographic map of target territories in the basal forebrain (*Figure 4I*).

## Progressive disruption of forebrain cholinergic axon arbor morphology in an AD model

As noted in the Introduction, loss of forebrain cholinergic innervation is a prominent feature of AD. To visualize AD pathology at the level of single cholinergic axon arbors, we used *APPswe/PS1ΔE9* doubly

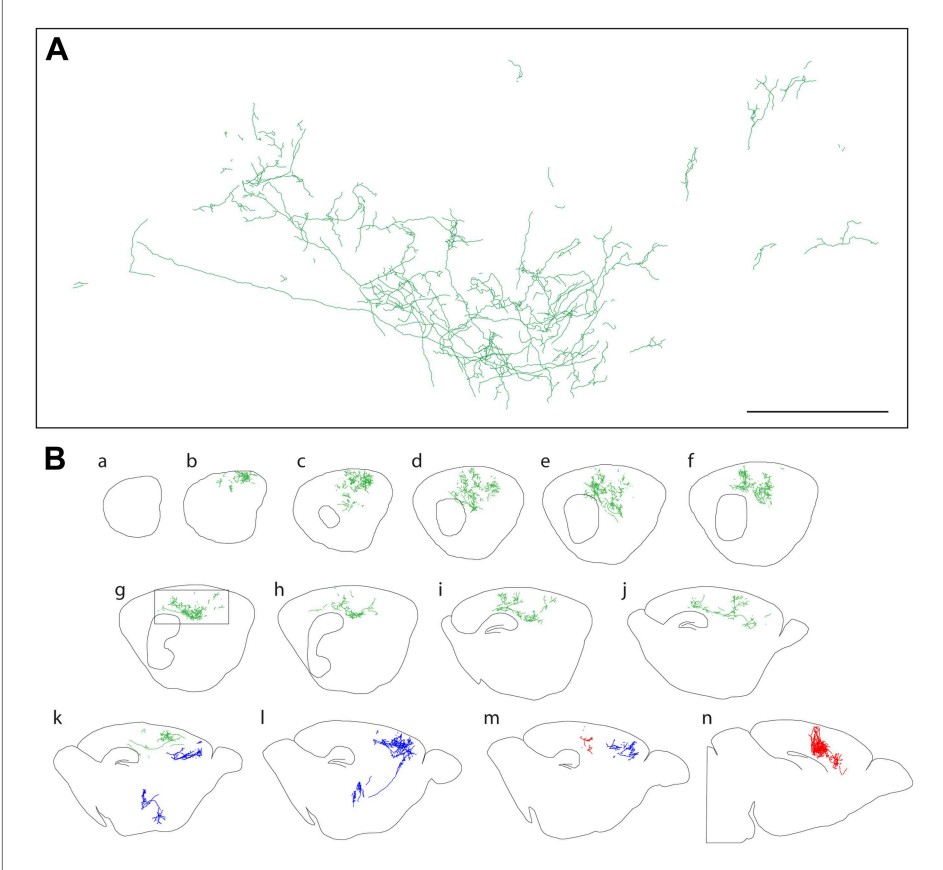

**Figure 3**. Axon arbors of forebrain cholinergic neurons from P12 *Chat-IRES-CreER;R26IAP* mice visualized with sparse Cre-mediated recombination. (**A**) Enlarged view of the boxed region from panel g in (**B**). Scale bar, 500 μm (corrected for tissue shrinkage in BBBA). (**B**) Fourteen consecutive 300 μm sagittal sections from a single P12 hemisphere (a–n) with three traced AP+ cholinergic neurons, colored red, green, and blue. The blue neuron is shown in its entirety, including the cell body and dendrites in the basal forebrain; for the green and red neurons, only the cortical axon arbors are shown.

transgenic mice (*Jankowsky et al., 2004*; referred to hereafter as *APP/PS1*) in which Aβ plaque rapidly accumulates with age in the cortex and hippocampus (*Figure 5—figure supplement 1*). This progression is accompanied by microglial reorganization and activation, and premature death (*Figure 5—figure supplement 1 and 2*). Analysis of AP+ cholinergic arbors in the cortex and hippocampus of 62 *Chat-IRES-CreER;R26IAP;APP/PS1* brains harvested between one and 12 months of age showed fragmentation of axons and a decrease in AP signal strength (*Figure 5A–C*), consistent with previously described changes in the appearance of ChAT immunoreactive fibers in mouse and human AD brains (*Figure 5—figure supplement 1C,D*; *Gordon et al., 2002*; *Schliebs and Arendt, 2011*). There was also a progressive accumulation of large numbers of AP+ and ChAT immunoreactive puncta with diameters up to ~10 μm in *Chat-IRES-CreER;R26IAP;APP/PS1* cortex and hippocampus but not in *Chat-IRES-CreER;R26IAP* controls (*Figure 5*, *Figure 5—figure supplement 1C,D*, *Figure 5—figure supplement 3A*; *Boncristiano et al., 2002*). These puncta presumably represent cholinergic axon breakdown products.

Bexarotene (Targretin) is a retinoid X receptor (RXR) agonist that has been reported to acutely promote ApoE-dependent clearance of soluble Aβ oligomers and to improve cognitive performance in the APP/PS1 mouse model of AD (*Cramer et al., 2012*). Whether bexarotene also promotes clearance of insoluble Aβ deposits (plaque) is controversial (*Fitz et al., 2013*; *Landreth et al., 2013*; *Price et al., 2013*; *Tesseur et al., 2013*; *Veeraraghavalu et al., 2013*). To determine whether bexarotene treatment alters the destruction of cholinergic axons in the APP/PS1 cortex, nine *Chat-IRES-CreER;R26IAP;APP/PS1* littermates were treated with 50–100 μg 4HT IP at P5 and were divided into

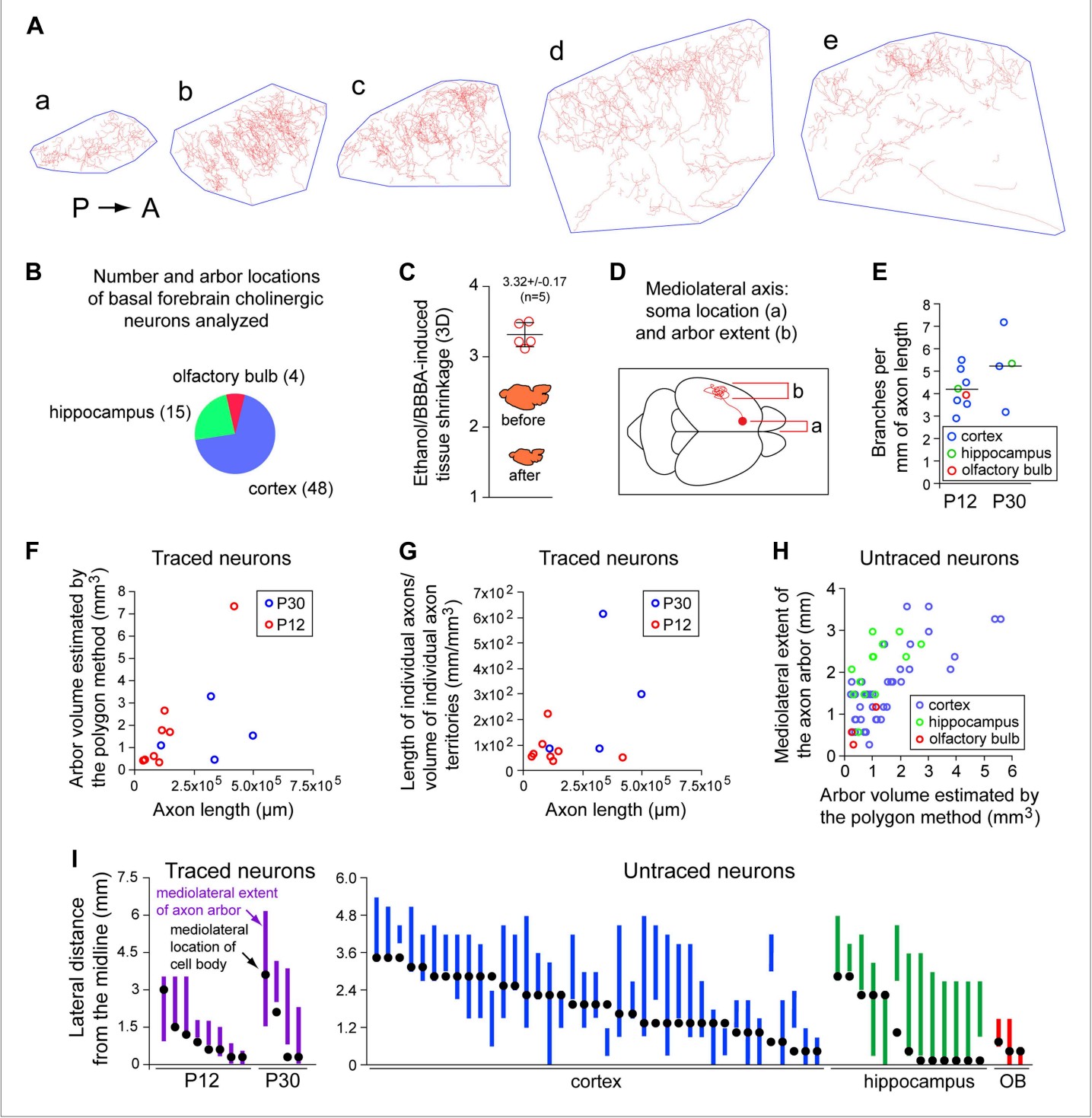

**Figure 4**. Quantitative analysis of morphologic parameters for cholinergic axon arbors. (**A**) The polygon method for estimating the target area for a single cholinergic axon arbor. Traced axon arbor images are shown for each 300 µm sagittal section from the P30 brain in *Figure 2C*, panels a–e. A minimal convex polygon has been drawn around each trace, providing an upper estimate of the cortical territory that is directly influenced by the arbor. As seen in panels d and e, the polygon method somewhat overestimates the target area by including regions that are relatively far from the axon. P, posterior. A, anterior. (**B**) Axon arbor locations for the 67 basal forebrain cholinergic neurons analyzed. (**C**) Quantification of tissue volume shrinkage due to dehydration in ethanol and BBBA. (**D**) Schematic of a forebrain cholinergic neuron in a dorsal view of the mouse brain showing the mediolateral cell body and arbor locations, the parameters displayed in panels (**H**) and (**I**). (**E**) The number of branch points per mm of axon length for the 12 forebrain cholinergic neurons that were traced. These data were obtained from two 300 µm sections per arbor by measuring the total axon length and counting

*Figure 4. Continued on next page*

*Figure 4. Continued*
all branch points for the AP+ arbor within each section. (**F**) Scatter plot of arbor volume (estimated using the polygon method) vs axon length for the 12 forebrain cholinergic neurons that were traced. (**G**) Scatter plot of axon density (length divided by arbor volume) vs axon length for the 12 forebrain cholinergic neurons that were traced. (**H**) Scatter plot of the mediolateral extent (defined in panel **D**) vs arbor volume (estimated using the polygon method) for the 55 forebrain cholinergic arbors that were not traced. (**I**) Mediolateral cell body and arbor locations for the 12 traced neurons (left) and the 55 untraced neurons (right). Black dots represent cell body location and the vertical bar represents the mediolateral extent of the axon arbor. OB, olfactory bulb.
The following figure supplements are available for figure 4:

**Figure supplement 1**. Cholinergic axon arbors: Z-stacked images, traces, and branch point locations.

three groups of three mice at 7 months of age: the first group received daily DMSO (vehicle) gavages for 14 days; the second group received daily 100 mg/kg bexarotene gavages for 14 days; and the third group received daily 100 mg/kg bexarotene gavages for 23 days, a regimen that activates RXR in the cerebral cortex as determined by the accumulation of ABCA1, a known RXR-inducible protein (*Figure 5—figure supplement 3B,C*; *Schmitz and Langmann, 2005*). As observed by others (*Fitz et al., 2013*; *Price et al., 2013*; *Tesseur et al., 2013*; *Veeraraghavalu et al., 2013*), the size and abundance of insoluble Aβ deposits and the number of activated microglia and astroglia around these deposits appeared to be unaltered by bexarotene treatment (data not shown), although we note that the DMSO formulation that we and Veeraraghavalu et al. used differs from the aqueous suspension of micronized bexarotene particles used by *Cramer et al. (2012)* (see also *Landreth et al., 2013*). Quantification of the density of AP+ puncta was also unaltered (*Figure 5—figure supplement 3D*), suggesting that the pathologic processes leading to the production of these puncta was not appreciably modified by the several week bexarotene treatment.

## Extremely large axon arbors in the mammalian brain: estimations and cross-species comparisons

The total length of the axons in a P30 mouse forebrain cholinergic neuron arbor–up to 50 cm–is roughly 25 times the linear dimension of the mouse brain. The large size of these arbors suggested that a systematic analysis of arbor sizes among various types of projection neurons might be of general interest (*Table 1*).

To the best of our knowledge there are only four studies (including the present one) in which individual axon arbors from the largest classes of CNS or PNS neurons have been traced and their lengths quantified. These are: (1) eight nigrostriatal dopaminergic neurons in the rat visualized following sparse infection with a GFP-expressing Sindbis virus (*Matsuda et al., 2009*), (2) a single CA3 pyramidal neuron in the rat visualized by neurobiotin injection (*Wittner et al., 2007*), (3) seven cutaneous sensory afferents of the 'large area, free-ending' class in mouse skin labeled by sparse CreER activation of an AP reporter (*Wu et al., 2012*), and (4) the four P30 mouse forebrain cholinergic neurons described here. The mean axon lengths for these four cell types were found to be, respectively: 47 cm (range: 14–78 cm), 48 cm, 71 cm (range: 64–98 cm), and 31 cm (range: 11–49 cm).

The validity of the single-cell axon length measurements for dopaminergic and cholinergic neurons can be independently checked with calculations based on the total volume of the target territory, the density of the particular type of axon (axon length per volume of target territory), and the number of neuronal cell bodies giving rise to that type of axon (*Table 1*). These population analyses are made possible by the availability of antibodies that localize to different types of axons: anti-ChAT for cholinergic axons (also visualized with acetylcholine esterase histochemistry), anti-tyrosine hydroxylase for striatal dopaminergic axons, and anti-serotonin for serotonergic axons. For example, *Anden et al. (1966)* estimated the total length of all dopaminergic axons in the rat striatum to be 7900 meters (bilaterally) and the number of mid-brain dopaminergic neurons projecting to the striatum to be 14,000 (bilaterally), giving a calculated mean axon length of 56 cm per dopaminergic neuron, in good agreement with the single cell tracing data of *Matsuda et al. (2009)* (*Table 1*).

A similar calculation can be performed for mouse forebrain cholinergic neurons using published estimates of the number of cell bodies in the nucleus basalis of Meynert (6632 and 4500), the volume of the mouse cortex as determined by MRI (109 mm$^3$), and the total length of cholinergic axons in the mouse cortex (1300 m), giving individual axon arbor length estimates of 20 cm and 29 cm (*Table 1*).

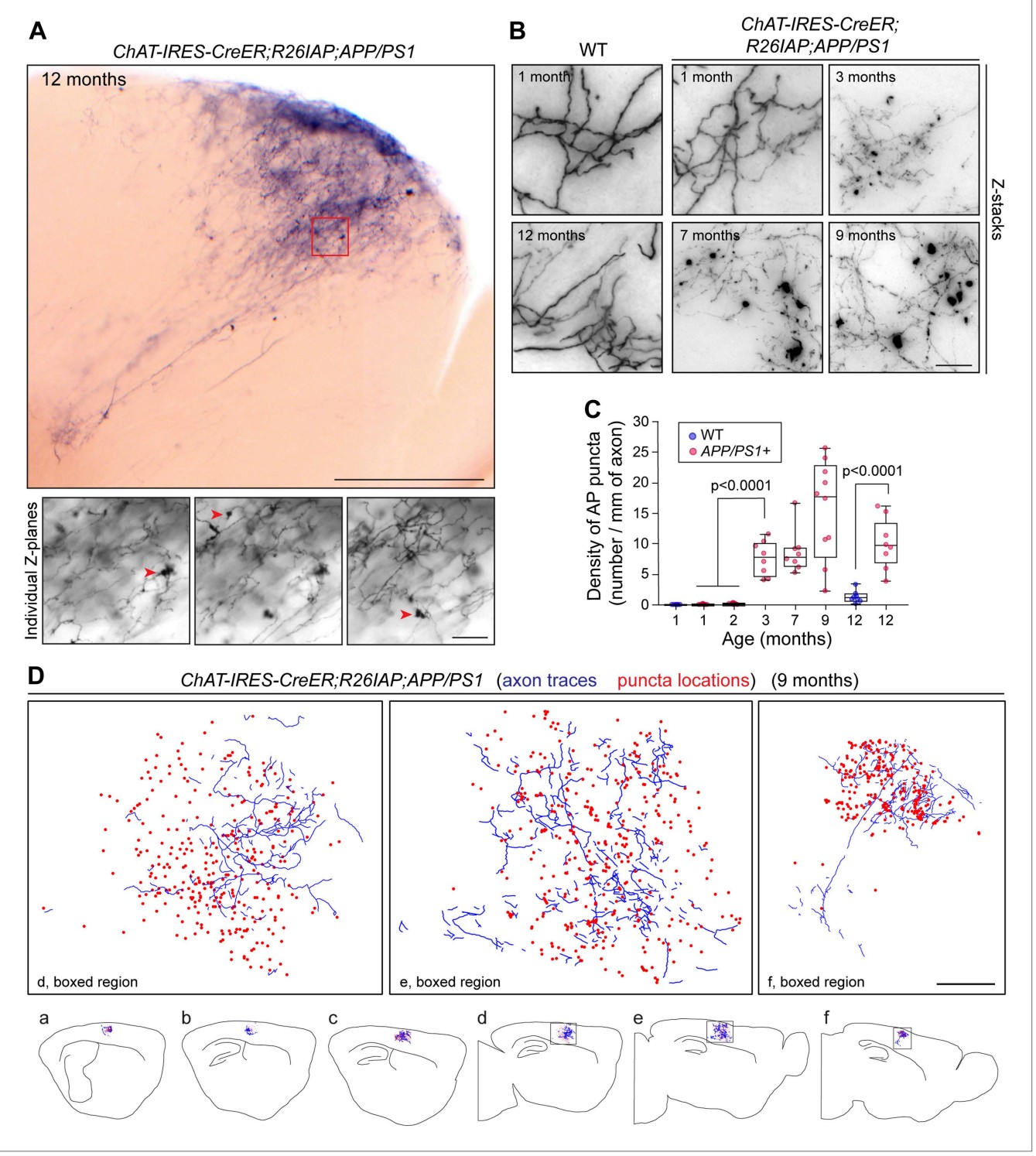

**Figure 5**. Disruption of cholinergic axon arbors in *Chat-IRES-CreER;R26IAP;APP/PS1* mice. (**A**) Upper panel, 300 µm sagittal section of a 12 month old *Chat-IRES-CreER;R26IAP;APP/PS1* brain showing part of a single AP+ axon arbor. The olfactory bulb is visible at lower right. Lower panels, three Z-planes enlarged from the region enclosed in the red square in the upper panel. Red arrowheads point to clumps of AP+ material (puncta). Scale bars: upper panel, 500 µm; lower panels, 50 µm. (**B**) Comparison of representative regions from forebrain cholinergic axon arbors in the cortex of *Chat-IRES-CreER;R26IAP* brains (WT; left) and *Chat-IRES-CreER;R26IAP;APP/PS1* brains (right), between one and 12 months of age. Structural heterogeneity, including the clumping of AP + material (puncta) and loss of AP staining intensity, increases with age in the *APP/PS1* background. Scale bar, 50 µm. (**C**) Quantification of AP+ puncta in the cortex and hippocampus of *Chat-IRES-CreER;R26IAP* (i.e., WT) and *Chat-IRES-CreER;R26IAP;APP/PS1* mice at
*Figure 5. Continued on next page*

*Figure 5. Continued*

different ages. Puncta appear at 3 months in *Chat-IRES-CreER;R26IAP;APP/PS1* mice. The box plots indicate the extreme data points (top and bottom bars), the 25–75% interval (box), and the median (central line). p-values, student's *t* test. (**D**) Complete tracing of an AP+ cortical cholinergic arbor (blue) with the locations of AP + puncta (red dots) indicated. Panels a–f show six adjacent 300 μm sagittal sections within which this arbor resides. The three enlarged images above correspond to the boxed regions in panels d–f. Scale bar, 500 μm (corrected for tissue shrinkage in BBBA).

The following figure supplements are available for figure 5:

**Figure supplement 1**. Aβ deposition, microglial reorganization, and disorganization of cholinergic fibers in the *APP/PS1* brain.

**Figure supplement 2**. Characterization of *APP/PS1* mice: survival and microglial activation.

**Figure supplement 3**. Tracing AP+ axons, and locating and quantifying AP+ puncta with and without bexarotene treatment.

(For these calculations we have omitted the volume of the hippocampus, which in mice represents 19% the volume of the cerebral cortex; its inclusion would modestly increase the estimates of axon length per neuron for those calculations based on axon density and cortical volume.) As a check on this calculation, we have independently measured the density of cholinergic axons in the mouse motor cortex (0.044 μm/μm$^3$; *Figure 6*) and also used the cholinergic axon density measured by *Descarries et al. (2005)* in parietal cortex (0.020 μm/μm$^3$) to calculate single cholinergic axon arbor lengths of 33–107 cm (*Table 1*). Given the limitations of the sampling methods, we consider these estimates of axon length to be in reasonably good agreement with the 11–49 cm range for the four P30 cholinergic neurons traced in the present study. An analogous calculation based on published data for rat forebrain cholinergic neurons gives a mean axon length of 62 cm (*Table 1*).

The estimates of axon length for large arbors in the rodent brain led us to ask whether it might be possible to perform analogous calculations for neurons in the human CNS. To the best of our knowledge, this simple calculation has not been performed previously. The human data for axon density and neuron counts have been published for forebrain cholinergic neurons and for serotonergic neurons projecting from the dorsal raphe nucleus to the cortex, and cortical volume estimates for humans are available from MRI analyses; forebrain cholinergic neuron data is also available for chimpanzees (*Table 1*). These calculations lead to axon length estimates of 107 m and 31 m, respectively, for human and chimpanzee forebrain cholinergic neurons, and an axon length estimate of 170–348 m for human serotonergic neurons. For both neurotransmitter systems, the vast majority of the visualized fibers within the cortex are presumed to derive from projection neurons because the density of cortical cell bodies labeled with anti-ChAT or anti-serotonin antibodies is extremely low (*Raghanti et al., 2008a*, *2008b*). Even if we allow for a possible under-estimate in neuron counts or a possible over-estimate in axon density measurements by as much as 2–3-fold, the calculations imply that in the human brain these two classes of projection neurons have axons that are, on average, many tens of meters in length.

## Discussion

The present work shows that mammalian forebrain cholinergic neurons are among the largest and most complex neurons described to date, as judged by total axon length and number of branch points. Nigrostriatal dopaminergic neurons appear to be of similar size and complexity, as are the largest cutaneous sensory neurons (*Table 1*, and references therein). The calculated total axon length for human forebrain cholinergic neurons, ~100 meters, is larger than the ~30 meter length estimated for blue whale corticospinal and DRG axons, which are typically cited as the 'largest' neurons in the animal kingdom (*Smith, 2009*). We note, however, that if one considers only the distance from the cell body to the tip of the most distal processes, then human forebrain cholinergic neurons are substantially 'smaller' in the sense of being more compact than many neurons with more nearly linear axons.

For forebrain cholinergic neurons, it is instructive to calculate the relative volumes and surface areas of soma and axon. If we take ~0.3 μm as the mean diameter of a rat cholinergic axon (*Vaucher and Hamel, 1995*; a value typical for unmyelinated CNS fibers; see; *Perge et al., 2012*), and temporarily neglect the volume added by varicosities, we obtain an axon volume of ~70.6 × 10$^3$ μm$^3$ per meter of axon length. Along the length of cholinergic fibers in the rat cortex and hippocampus, varicosities occur at a mean frequency of ~0.4/μm and their mean internal volume is ~0.06 μm$^3$ (the mean diameter is 0.48 μm; *Mechawar et al., 2000*, *2002*; *Aznavour et al., 2005*; *Descarries et al., 2005*). Thus, the varicosities would add a

**Table 1.** Axon arbor lengths for diffuse projection neurons

**Basal forebrain cholinergic neurons (nucleus basalis of Meynert to cortex)**

| Species | Number of neurons | Axon density in cortex | Axon length in cortex | Cortical volume | Mean axon length/neuron |
|---|---|---|---|---|---|
| Mouse | 6632 (a) | | 1300 m (b) | 109 mm³ (c) | 20 cm |
| | 4500 (b) | | 1300 m (b) | 109 mm³ (c) | 29 cm |
| Mouse | 6632 (a) | 0.020–0.044 μm/μm³ (d; this study) | | 109 mm³ (c) | 33–72 cm |
| | 4500 (b) | 0.020–0.044 μm/μm³ (d; this study) | | 109 mm³ (c) | 48-107 cm |
| Mouse* | 4 traced neurons following CreER/loxP labeling (this study) | | | | 31 cm |
| Rat | 7,312 (d) | 0.0113 μm/μm³ (e) | | 400 mm³ (f) | 62 cm |
| Chimp | 315,000 (g) | 0.066 μm/μm³ (h) | | 147 cm³ (i) | 31 m |
| Human | 435,000 (g) | 0.080 μm/μm³ (h) | | 583 cm³ (i) | 107 m |

**Nigro-striatal dopaminergic neurons**

| Species | Number of neurons | Varicosities per axon length | Number of TH + varicosities in the striatum | Mean axon length/neuron |
|---|---|---|---|---|
| Rat | 7000 (j) | 5–7 varicosities/7 μm (j) | $3.4 \times 10^9$ per side (j) | 55–77 cm |
| Rat* | 8 traced neurons following sparse GFP virus infection (k) | | | 47 cm |

**Serotonergic neurons (dorsal raphe nucleus to cortex)**

| Species | Number of neurons | Axon density in cortex | Cortical volume | mean axon length/neuron |
|---|---|---|---|---|
| Rat | 11,500 (l) | 0.023 μm/μm³ (m) | 400 mm³ (f) | 80 cm |
| | 15,191 (n) | 0.023 μm/μm³ (m) | 400 mm³ (f) | 61 cm |
| Human | 80,386 (o) | 0.048 μm/μm³ (r) | 583 cm³ (i) | 348 m |
| | 86,565 (p) | 0.048 μm/μm³ (r) | 583 cm³ (i) | 323 m |
| | 165,000 (q) | 0.048 μm/μm³ (r) | 583 cm³ (i) | 170 m |

**Hippocampal CA3 pyramidal neurons**

| Rat* | 1 traced neuron following neurobiotin injection (s) | 48 cm |
|---|---|---|

**Cutaneous sensory neurons with free endings in back skin**

| Mouse* | 7 traced neurons following CreER/loxP labeling (t) | 71 cm |
|---|---|---|

References: (a) *Perez et al., 2007*; (b) *Boncristiano et al., 2002*; (c) *Kovacević et al., 2005*; (d) *Miettinen et al., 2002*; (e) *Mechawar et al., 2000*; (f) *Mengler et al., 2013*; (g) *Raghanti et al., 2011*; (h) *Raghanti et al., 2008a*; (i) *Rilling and Insel, 1999*; (j) *Anden et al., 1966*; (k) *Matsuda et al., 2009*; (l) *Descarries et al., 1982*; (m) *Cunningham et al., 2005*; (n) *Vertes and Crane, 1997*; (o) *Underwood et al., 2007*; (p) *Underwood et al., 1999*; (q) *Baker et al., 1991*;(r) *Raghanti et al., 2008b*; (s) *Wittner et al., 2007*; (t) *Wu et al., 2012*. The asterisk (*) marks experiments in which individual axon arbors were traced.

volume of $(0.4 \times 10^6) \times (0.06 \ \mu m^3) = 24 \times 10^3 \ \mu m^3$ per meter of axon length. The mean soma volume for rat basal forebrain cholinergic neurons is ~$14 \times 10^3 \ \mu m^3$ (diameters are 18–43 μm; the mean is taken as ~30 μm; *Butcher, 1995*). Using the value of 62 cm calculated here for the mean length of rat forebrain cholinergic axons (*Table 1*), the axon and varicosity volumes ($44 \times 10^3 \ \mu m$ and $15 \times 10^3 \ \mu m^3$) sum to ~$59 \times 10^3 \ \mu m^3$, or ~4.2 times greater than the soma volume. Thus, despite its great length, the axon arbor only increases the neuron's volume several-fold. By contrast, the calculated mean surface area of the axon arbor (~$5.84 \times 10^5 \ \mu m^2$) is ~210 times greater than the calculated mean surface area of the soma (~$2800 \ \mu m^2$).

To appreciate the relative dimensions of a typical forebrain cholinergic neuron, the reader may find it useful to consider a model in which this neuron has been enlarged by a factor of $10^4$ (*Figure 7*).

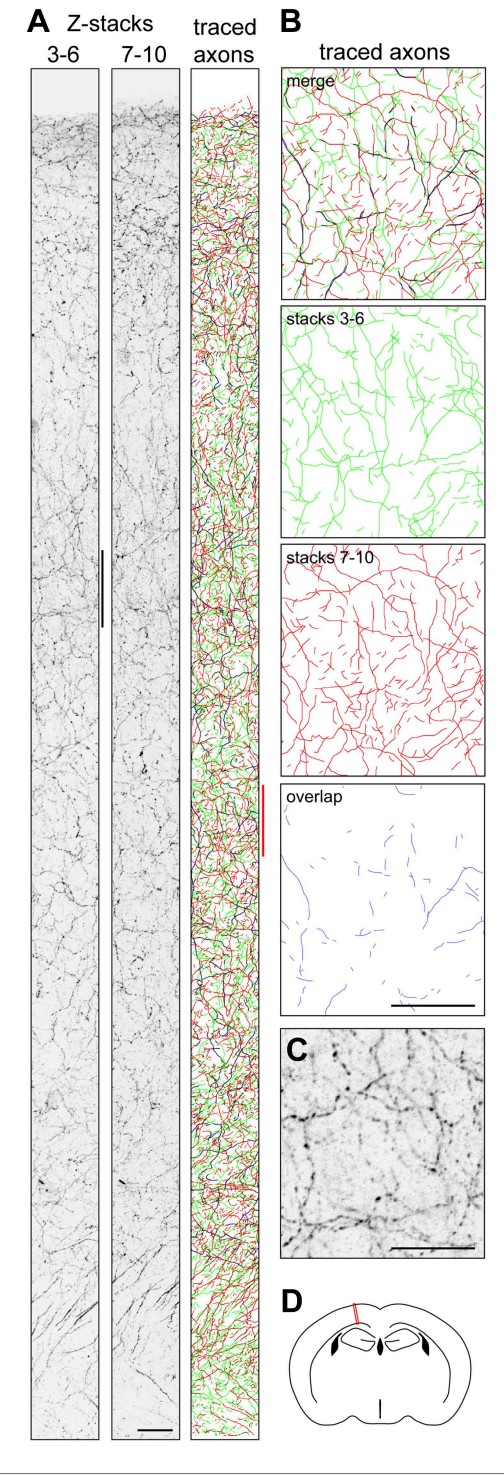

**Figure 6**. Quantifying ChAT+ axon density in P30 mouse cortex. (**A**) Coronal section of P30 mouse motor cortex following ChAT immunostaining. The cortical surface is at the top; the base of the cortex is at the bottom. Confocal images of the fluorescently immunostained tissue (converted to grey scale and inverted) were captured at Z-plane separations of 2 μm. Z-stacks

*Figure 6. Continued on next page*

(For this model, we use the data on rat cholinergic neurons from the preceding paragraph.) The model neuron would have a cell body ~30 cm in diameter, a dendritic arbor extending as far as ~5 meters from the soma, and axons of ~3 mm diameter divided into a proximal segment ~200 meters in length connected to a highly branched distal arbor with a total length of 6 kilometers (!). For the human counterpart, the model would have a total axon length of ~1000 kilometers.

## Forebrain cholinergic neuron morphology and function

The large sizes of the cholinergic axon arbors described here are consistent with a spatially diffuse modulatory role for cholinergic transmission in the cortex (*Descarries et al., 1997*; *Lucas-Meunier et al., 2003*). In the mouse, cortical columns are approximately 150–300 μm across, the diameter of an individual barrel in the barrel cortex (*Jan et al., 2008*). Since the typical cholinergic axon arbor extends over two millimeters in the plane of the cortex (*Figures 2 and 3*), each arbor contacts multiple cortical columns. Although locally induced or locally restricted acetylcholine release is possible, it seems likely that signals originating in the dendrites and leading to action potentials at the soma of a forebrain cholinergic neuron would affect synaptic output over the entire arbor. This line of reasoning implies that cholinergic modulation of cortical and hippocampal function in response to information originating in the basal forebrain is likely to have low spatial resolution. A similar argument can be applied to nigrostriatal dopaminergic signaling (*Matsuda et al., 2009*).

## Implications of extremely large axon arbors for neurodegenerative disease

The very large axon lengths calculated for human forebrain cholinergic neurons reflect (1) the enormous evolutionary expansion of the human cerebral cortex, which is ~5000 times larger than the mouse cerebral cortex, (2) a more modest expansion in the number of basal forebrain cholinergic neurons, which differ between humans and mice by only a factor of 60–100, and (3) a ~twofold higher density of cholinergic axons in the human cortex (*Table 1*). The same line of reasoning applies to the rodent vs human difference in the size of individual serotonergic axon arbors (*Table 1*).

Extremely large axon arbors present a cell biological challenge, as they require a correspondingly large expenditure of resources for growth, maintenance, and repair, especially as related to membrane synthesis and axonal transport. It is

*Figure 6. Continued*

encompassing planes 3–6 and 7–10 are shown. The traced axons for both sets of Z-stacks are color-coded with axons in stacks 3–6 in green, axons in stacks 7–10 in red, and regions of overlap in blue. Scale bar, 25 µm. (**B**) Axon tracings corresponding to the region in the right panel in (**A**) that is demarcated by the vertical red line. Scale bar, 25 µm. (**C**) ChAT immunostaining corresponding to the region adjacent to the left panel in (**A**) that is demarcated by the vertical black line. Scale bar, 25 µm. (**D**) The red rectangle shows the region of motor cortex analyzed in (**A**), at approximately Bregma −1.06.

possible that the vulnerability of forebrain cholinergic neurons in the context of AD is related, at least in part, to the large size of their axon arbors. For example, vulnerability might be related to a particular sensitivity of the axonal transport machinery to biochemical perturbations associated with Aβ toxicity. Perhaps more significantly, the requirement that all transportation processes between cell body and axon arbor funnel through a single proximal axon segment suggests that trafficking within this segment may limit the efficacy of cellular responses to axonal damage or stress. The general idea that extreme axon length increases vulnerability to neurodegeneration has been discussed in the context of motor neuron disease (*Cavanagh, 1984*; *Ferraiuolo et al., 2011*), and it seems reasonable that this concept might apply to a wide variety of neurons with very large axon arbors.

## Materials and methods

### Mouse lines and neuronal sparse labeling

Experiments unrelated to AD were performed with *Chat-IRES-CreER/+;R26IAP/+* mice (referred to in the text as *Chat-IRES-CreER;R26IAP*). For AD experiments, *R26IAP/R26IAP* mice were crossed to *Chat-IRES-CreER/Chat-IRES-CreER;APP/PS1/+* mice to obtain *R26IAP/+;Chat-IRES-CreER/+* (WT control) and *R26IAP/+;Chat-IRES-CreER/+;APP/PS1/+* littermates. The *APPswe/PS1ΔE9* line was a gift from Dr Phil Wong (Johns Hopkins University). Cholinergic neuron labeling followed intraperitoneal (IP) delivery of 1–5 µg 4-hydroxytamoxifen (4HT) at P4-P5 with analysis at P12, P30 and later ages, as indicated. A wide range of 4HT doses–from 1 µg to 400 µg– was tested in an initial survey of ~250 mice to identify the optimal dose for sparse labeling. Mice were handled in accordance with the Institutional Animal Care and Use Committee (IACUC) guidelines of the Johns Hopkins Medical Institutions.

### AP histochemistry

Mice were deeply anesthetized with ketamine/xylazine and then sacrificed by trans-cardiac perfusion with neutral buffered 10% formalin solution (Sigma-Aldrich, St. Louis, MO; equivalent to 4% paraformaldehyde). Brains or eyes were heated to 70°C for 90 min to inactivate endogenous phosphatase activity. Serial brain sections of 300 µm thickness were produced with a VT1200 vibratome (Leica, Buffalo Grove, IL). AP histochemistry and clearing in 2:1 benzyl benzoate:benzyl alcohol (BBBA) were performed as described (*Wu et al., 2012*). For long-term storage, AP-stained brain sections were equilibrated in ethanol and stored at −20°C.

### Immunofluorescence and confocal imaging

The following antibodies were used for immunostaining of 50–100 µm thick floating brain sections: goat anti-ChAT, 1:1000 (AB143; Millipore, Billerica, MA); rabbit anti-GFP, 1:1000 (A11122;

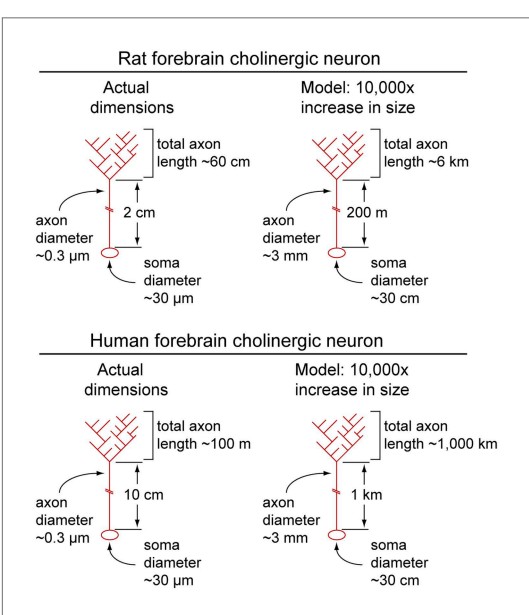

**Figure 7**. Calculated dimensions of rat and human forebrain cholinergic neurons. Actual dimensions of human and rat forebrain cholinergic neurons, calculated from the data in *Table 1* (left image in each pair). A macroscopic model in which the linear dimensions of each cholinergic neuron have been multiplied 10,000-fold (right image in each pair). Soma and axons are shown; dendrites have been omitted.

Invitrogen); mouse anti-β-amyloid (6E10), 1:500 (NE1003; Millipore); rabbit anti-GFAP, 1:1000 (AB5804; Millipore); rabbit anti-Iba1, 1:1000 (019-19741; Wako, Richmond, VA); mouse anti-GFAP 1:1000 (MAB360; Millipore); mouse anti-ABCA1 monoclonal antibody HJ1 (ab66217; Abcam, Cambridge, MA); mouse anti-β-Actin antibody AC-15 (A5441; Sigma); and rabbit anti-TH, 1:1000 (AB152; Millipore). Secondary antibodies were from Invitrogen. GS-lectin staining used Alexa488-IB4, 1:1000 (I21411; Invitrogen, Grand Island, NY). Brain sections were incubated in primary antibodies diluted in PBS, 0.5% Triton X-100, 0.1 mM CaCl$_2$ (PBSTC) + 10% normal goat or donkey serum, washed in PBSTC for 6 hr, and incubated at 4°C overnight in secondary antibodies diluted in PBSTC +10% normal goat or donkey serum. After washing in PBSTC for 4–6 hr, brain sections were mounted in Fluoromount G (17984-25; EM Sciences, Hatfield, PA). Images were captured on a Zeiss LSM700 confocal microscope and processed with Zen software, ImageJ/Fiji, and Adobe Photoshop.

## AP image analysis

Only brains with fewer than five AP+ neurons/hemisphere were subjected to detailed analysis. For high-resolution analyses, isolated arbors were imaged in bright-field mode at 10X magnification with Z-planes separated by 3 µm. Grey-scale images were captured with a Zeiss Imager Z1 system in montage mode and assembled with Zeiss AxioVision software. Neurites were traced using Neuromantic neuronal tracing freeware (Darren Myat, http://www.reading.ac.uk/neuromantic) in semi-automatic mode as described (*Wu et al., 2012*). The total length of an axon arbor was obtained by summing up the lengths of the traces derived from that arbor within each tissue section. (Following dehydration in ethanol and equilibration in BBBA, 300 µm thick vibratome sections of brain undergo an isotropic volume shrinkage of 3.32+/−0.17 [*Figure 4C*]. Axon lengths reported here have been corrected for that shrinkage.) Branch points were manually scored using ImageJ/Fiji. To estimate arbor volumes, the smallest convex polygon encompassing all axon segments for a given arbor in the Z-stacked image for each 300 µm section was drawn over the image using ImageJ/Fiji, and each polygon area was calculated. Sections that contained only the subcortical axon segment were not included in the polygon analysis. Statistical analyses were performed with Excel and Graph-Pad. Error bars in the figures indicate standard deviation (SD). p-values were calculated with the student's *t* test.

## Quantification of AP puncta

Eight randomly selected 400 µm × 400 µm images of WT or *APP/PS1+* axon arbors from each of three mice per genotype and per time point, with or without bexarotene treatments, were acquired with the Zeiss Imager Z1 system as described above. AP deposits >5 pixels in diameter (images were 620 × 620 pixels) were manually counted using ImageJ/Fiji software. No AP axon fragmentation or AP deposits were detected in WT mice younger than 8 months.

## Quantification of ChAT immunostained axons in motor cortex

Confocal images at Z-plane intervals of 2 µm within the interior of a 50 µm thick vibratome section were combined in two adjacent Z-stacks of four planes each (i.e., 8 µm thickness per Z-stack) and ChAT+ axon segments were traced using Neuromantic software. To estimate the length of the traced axon segments residing outside of each Z-stack–an artifact that results from the incomplete elimination of out-of-plane signals in the confocal image–the total length of those axon segments that were traced from both of the adjacent Z-stacks was measured, and found to comprise 25.4% of the total axon length traced for each Z-stack. As half of this overlap derives from each Z-stack and as this effect occurs on both surfaces of each Z-stack, the corrected axon length for each Z-stack was calculated by subtracting 25.4% from the total length of the trace from each Z-stack.

## Bexarotene treatment of aged *APP/PS1+* mice

Bexarotene was purchased from Sigma-Aldrich (SML0282) and dissolved in DMSO. Mice received daily gavages of 100 µl DMSO with or without bexarotene. Nine *Chat-IRES-CreER;R26IAP;APP/PS1* littermates at 7 months of age were divided into three groups of three mice per group: group 1 received vehicle only (DMSO) daily for 14 days; group two received 100 mg/kg bexarotene daily for 14 days; and group three received 100 mg/kg bexarotene daily for 23 days. All mice received 50–100 µg 4HT IP at P5. Mice were perfused and analysed by AP histochemistry and immunofluorescent staining as described above.

For ABCA1 immunoblotting, 6 mice at 7 months of age were divided into two groups: 3 mice received vehicle only (DMSO) daily for 7 days, and the 3 mice received 100 mg/kg bexarotene daily for

7 days. On the eighth day, the cerebral cortices were homogenized in ~1 ml PBS with 0.5% Triton X-100, 1 mM phenylmethylsulfonyl fluoride (PMSF), and complete protease inhibitor cocktail tablet (11697498001, Roche, Indianapolis, IN), to give a total lysate protein concentration of ~18 µg/µl. Proteins in SDS sample buffer were loaded without boiling onto a 7.5% SDS/polacrylamide gel and immunoblotted with mouse anti-ABCA1 monoclonal antibody HJ1 (ab66217; Abcam). Conveniently, native mouse IgG in the extract runs at lower molecular weight than ABCA1. Monoclonal Anti-β-Actin antibody AC-15 (A5441; Sigma) was used as a loading control.

## Acknowledgements

The authors thank Drs Glenda Halliday and Mary Ann Raghanti for advice, Dr Philip Wong for the *APP/PS1* mice, and Lucas Hua, Dr Amir Rattner, Yulian Zhou, and two reviewers for helpful comments on the manuscript. Supported by the Human Frontiers Science Program, the Howard Hughes Medical Institute, and the Brain Sciences Institute of the Johns Hopkins University.

## Additional information

### Competing interests

JN: Reviewing editor, *eLife*. The other authors declare that no competing interests exist.

### Funding

| Funder | Grant reference number | Author |
|---|---|---|
| Human Frontier Science Program | LT000125/2009-L | Hao Wu |
| Howard Hughes Medical Institute | | Hao Wu, John Williams, Jeremy Nathans |
| Brain Sciences Institute of the Johns Hopkins University | | Jeremy Nathans |

The funders had no role in study design, data collection and interpretation, or the decision to submit the work for publication.

### Author contributions

HW, JN, Conception and design, Acquisition of data, Analysis and interpretation of data, Drafting or revising the article, Contributed unpublished essential data or reagents; JW, Acquisition of data, Analysis and interpretation of data

### Ethics

Animal experimentation: This study was performed in strict accordance with the recommendations in the Guide for the Care and Use of Laboratory Animals of the National Institutes of Health. All of the animals were handled according to approved institutional animal care and use committee (IACUC) protocol MO13M469 of the Johns Hopkins Medical Institutions.

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
