## [Decision Letter]

Thank you for sending your work entitled “Complete morphologies of basal forebrain cholinergic neurons in the mouse” for consideration at *eLife*. Your article has been favorably evaluated by a Senior editor and 2 reviewers, one of whom, Franck Polleux, is a member of our Board of Reviewing Editors.

The Reviewing editor and the other reviewer discussed their comments before we reached this decision, and the Reviewing editor has assembled the following comments to help you prepare a revised submission.

Overall, the reviewers were very enthusiastic about the results, the methodology, and the general conclusions of your study. We would like you to address the minor comments listed below that will involve rather small changes in the data presentation and/or text editing. The only main concern listed in the minor comments of Reviewer#1 that we would like you to address concerns the bexarotene data. In particular, you should decide to either (1) provide evidence for bioavailability of your bexarotene treatment (in order to exclude the possibility that your negative results are due to ineffective regimen), or (2) you can decide to remove this data altogether, which we don't think would decrease the main take-home message of the paper. It's your decision.

Please let us know your decision and submit your revised manuscript accordingly with a cover letter detailing the changes you made.

Main issues to address:

Reviewer #1:

1) The only real unknown in the paper concerns the effectiveness of the bexarotene treatment. Because of the negative nature of the results obtained by the authors regarding the effect on axon maintenance in the mouse AD model, the authors need to provide some analysis of the effectiveness of the bexarotene treatment which is provided by oral gavage. Is the dose (solubility?) and regimen used here similar to the original paper by Cramer et al? The authors should provide some measurement of the amount of bexarotene that crosses the BBB or a positive control (biochemical?) of the effectiveness of the treatment since the negative results could simply be due to ineffective dosage and/or brain accessibility compared to the original paper. This is actually an important issue since the present results would further disprove this original study.

Reviewer #2:

2) The use of supplemental figures that have their own independent numbering system that is linked to a main Figure is cumbersome and needlessly complicating. Indeed, as this is presumably an on-line only publication, page space is clearly not a concern. Please eliminate all the supplemental figures, incorporating them into the text as a linear story, which is how you present it all. The bexarotene study, a negative result, could be deleted altogether (it could be described sufficiently in the text).

3) In the Results section, the authors refer to Figure 4, but this figure does not appear to illustrate this. Perhaps I have misunderstood, but the only 2 mm mentioned in this paragraph that might relate to this Figure is the 2 mm (?) mean breadth of the axon arbor in the M-L dimension.

4) The final Figure 7 includes an example of a rat forebrain neuron and a human forebrain neuron (why not a mouse?); plus the same dimensions scaled up 10,000 times. I found the scaling exercise largely without benefit, and believe the entire figure could probably be eliminated (certainly the right half of each panel), with the discussion text just indicating these relative magnitudes of the real cells, for instance, the total axonal length relative to the somal diameter, or to the axonal diameter, etc. If you do maintain any of Figure 7, the micron symbol should be employed rather than a “u”.

5) As you have the densities of the neurons, are you in a position to comment on the degree of dendritic servitude per unit volume in the basal forebrain, and related to this, can you assess/comment on the extent to which the axonal territories overlap at all? Perhaps you have informative data from occasionally closer pairs of labeled cells that might lend support to calculations based on cell number, average arbor size, and cortical volume, etc.

---

## [Author Response]

We would prefer to keep the bexarotene data in the paper. It is a minor (and negative) part of the study. While we concede that it is not central to the paper, we think that it should be in the public domain.

Reviewer #1:

*1) The only real unknown in the paper concerns the effectiveness of the bexarotene treatment. Because of the negative nature of the results obtained by the authors regarding the effect on axon maintenance in the mouse AD model, the authors need to provide some analysis of the effectiveness of the bexarotene treatment which is provided by oral gavage. Is the dose (solubility?) and regimen used here similar to the original paper by Cramer et al? The authors should provide some measurement of the amount of bexarotene that crosses the BBB or a positive control (biochemical?) of the effectiveness of the treatment since the negative results could simply be due to ineffective dosage and/or brain accessibility compared to the original paper. This is actually an important issue since the present results would further disprove this original study*.

That is an excellent suggestion. We have fleshed out this issue in the Results section and in an expanded version of Figure 5—figure supplement 3 (panels B and C). In our experiments, we delivered bexarotene dissolved in DMSO by daily oral gavage, which is the regimen used by Veeraraghavalu et al. (2013 Science 340: 924-f), one of three research groups that attempted unsuccessfully to reproduce the findings of [12]. In their response to the letters from the four groups that failed to reproduce their findings, the authors of [12] noted that they had used a micronized formulation of bexarotene and suggested that differences in pharmacokinetics may have accounted for the discrepant findings ([26] Science 340: 924g). We have not tried to reproduce the data of Cramer et al., so our data does not directly contradict their findings. However, our bexarotene experiment was inspired by their report, so it would be appropriate for us to note that our drug formulation differed from theirs and that the authors of Cramer et al have suggested that such differences could be biologically significant. We have now added a sentence to this effect in the Results section.

We have also added data from an experimental test of bexarotene’s CNS bioavailability using our drug regimen. ABCA1 is a transcriptional target of RXR signaling and [53] used the increase in ABCA1 protein (monitored by anti-ABCA1 immunoblotting) to assess bexarotene activation of RXR in the brain. We have conducted this type of analysis (using the same mAb used by Veeraraghavalu et al.) to assess the effectiveness of bexarotene treatment in activating RXR in the CNS. Specifically, we compared ABCA1 levels in three vehicle-treated and three bexarotene-treated mice at 7 months of age, using the same drug administration protocol that we used for the cholinergic neuron analysis in AD model mice. The result is a robust induction of ABCA1 in the brain cortex in response to one week of bexarotene treatment, in line with what Veeraraghavalu et al. observed. This new data is now included in Figure 5—figure supplement 3 (panels B and C).

Reviewer #2:

*2) The use of supplemental figures that have their own independent numbering system that is linked to a main Figure is cumbersome and needlessly complicating. Indeed, as this is presumably an on-line only publication, page space is clearly not a concern. Please eliminate all the supplemental figures, incorporating them into the text as a linear story, which is how you present it all. The bexarotene study, a negative result, could be deleted altogether (it could be described sufficiently in the text)*.

The general comment about supplemental figures should probably be directed to the editors of *eLife*. We think this division into main figures and supplemental figures serves the purpose of permitting data that is supportive but not central to be placed in the public domain. In the present case, the analysis of dendrite length and the immunohistochemical characterization of the AD mouse brain fall into the latter category. We also note that in the final version of the paper on the *eLife* web site, it is easy to click back and forth between main figures and supplemental figures. We would prefer to keep the bexarotene data in the paper. It is a minor (and negative) part of the study. While we concede that it is not central to the paper, we think that it should be in the public domain.

*3) In the Results section, the authors refer to*
Figure 4*, but this figure does not appear to illustrate this. Perhaps I have misunderstood, but the only 2 mm mentioned in this paragraph that might relate to this Figure is the 2 mm (?) mean breadth of the axon arbor in the M-L dimension*.

Yes, the comment referring to “2 mm” does refer to the mean lateral breadth of the axon arbors. This is an example of a linear dimension that illustrates the point we were trying – perhaps unsuccessfully – to make. We have now clarified the writing.

*4) The final*
Figure 7
*includes an example of a rat forebrain neuron and a human forebrain neuron (why not a mouse?); plus the same dimensions scaled up 10,000 times. I found the scaling exercise largely without benefit, and believe the entire figure could probably be eliminated (certainly the right half of each panel), with the discussion text just indicating these relative magnitudes of the real cells, for instance, the total axonal length relative to the somal diameter, or to the axonal diameter, etc. If you do maintain any of*
Figure 7*, the micron symbol should be employed rather than a “u”*.

We have corrected “u” and changed it to the Greek letter “µ”. We use the rat rather than mouse to illustrate and discuss the dimensions of cholinergic neurons (both in the text and in Figure 7) because there is EM data on rat cholinergic neurons in the scientific literature. The 10,000x scale model presented in Figure 7 is a useful didactic tool. We have found that most people who see this data – including neurobiologists – don’t fully appreciate the extraordinary dimensions of these cells until those dimensions are converted to the scale of everyday experience.

*5) As you have the densities of the neurons, are you in a position to comment on the degree of dendritic servitude per unit volume in the basal forebrain, and related to this, can you assess/comment on the extent to which the axonal territories overlap at all? Perhaps you have informative data from occasionally closer pairs of labeled cells that might lend support to calculations based on cell number, average arbor size, and cortical volume, etc*.

These are excellent suggestions. We do not have a useful data set of overlapping axon arbors, but we have population data from which these parameters can be calculated. We have now done those calculations and they are now included in the Results section. The calculated redundancy of coverage of forebrain cholinergic axon arbors in the cortex and hippocampus is between 47 and 69, that is on average each point in the cortex and hippocampus resides within the arbor territories of 47–69 forebrain cholinergic arbors. The analogous calculation for dendritic territories within the basal forebrain yields a coverage factor of 470–690. The length density of cholinergic dendrites in the basal forebrain is 0.078–0.114 um/cubic um.

With respect to the calculations of the surface and volume of cholinergic neurons in the Discussion section and the diagram in Figure 7, in the revised manuscript we have changed the estimate of unmyelinated axon diameter from 0.2 um to 0.3 um, a value that more closely approximates the diameters of typical unmyelinated CNS axons (37). We have also corrected two typographical errors in the description of the surface area ratio of soma and axon arbor, and a typographical error in Figure 7.